# A Measurement Scale to Assess Children’s Satisfaction with Hospitalization in the Andalusian Population

**DOI:** 10.3390/ijerph16173110

**Published:** 2019-08-27

**Authors:** Montserrat Gómez-de-Terreros-Guardiola, José F. Lozano-Oyola, María-Dolores Lanzarote-Fernández, Raquel Rupérez Bautista, Isabel Avilés-Carvajal, Tonio Schoenfelder, Rafael J. Martínez-Cervantes

**Affiliations:** 1Department of Personality, Assessment and Psychological Treatments, University of Seville, 41018 Seville, Spain; 2Institute for Public Health, School of Medicine, Technical University of Dresden, 01307 Dresden, Germany; 3Department of Experimental Psychology, University of Seville, 41018 Seville, Spain

**Keywords:** patient satisfaction, satisfaction scale, hospitalization, children and teenagers, internal consistency, exploratory factor analysis, reliability

## Abstract

*Background*: Patient satisfaction is a principal indicator in the evaluation of the stay of pediatric patients in hospitals, since its consequences can emotionally interfere with health treatment. The aim of this study was to obtain a valid scale to assess children’s satisfaction with their time spent as a patient in an Andalusian hospital. *Method*: The Children’s Satisfaction with Hospitalization Questionnaire (CSHQ) was applied to 623 pediatric patients hospitalized in Andalusia. An exploratory factor analysis (EFA) showed one dimension underlying the children’s satisfaction with their hospitalization. After that, we developed a depuration analysis process to achieve a valid and unidimensional scale to assess children’s satisfaction. *Results*: The eleven-item one-dimension solution showed suitable consistency and goodness-of-fit indices. The final scale addresses hosting aspects as the main dimension of a minor’s satisfaction in Andalusian hospitals. *Conclusion*: A unidimensional scale has been determined for the assessment of children’s satisfaction with their stay in Andalusian hospitals based on hosting aspects. Nonetheless, other dimensions underlying the satisfaction of patients should also be considered.

## 1. Introduction

There is no globally accepted formulation of patient satisfaction [1]. However, satisfaction has been defined as a complex emotion that occurs in the interaction between expectations about a given situation and the subsequent perception of what has occurred. It has also been considered as a multidimensional construct that takes multiple dimensions into account [2,3,4,5].

Patient satisfaction is a principal indicator in evaluating the effectiveness of medical care during hospitalization [6]. Therefore, the assessment of satisfaction with hospitalization in pediatric patients is also relevant, since the consequences of this experience can be highly detrimental. In general, children feel fear and anxiety towards the hospital and medical procedures [7], which may interfere with the effectiveness of health treatment. Furthermore, satisfaction is an indicator of the quality of hospital service [8], which is a constant concern of all personnel involved in the care process [9]. 

It is therefore relevant to evaluate the children’s opinion about their own satisfaction with hospitalization, since it enables the point of view of the direct recipients of care to be ascertained by identifying any distress they may have experienced. This kind of data may enable the quality of service and health planning to be improved [10,11,12,13]. However, the opinion of the children has seldom been taken into account, since parents or guardians remain the main agents of information [14,15,16,17,18,19,20,21]. Nonetheless, there is an emerging trend whereby children and adolescents are directly asked about this issue [22,23,24,25,26,27].

Studies have also been conducted with satisfaction questionnaires that included children and adolescents in their samples [24,28,29], but they are neither directly oriented towards pediatric patients nor are they intended for general hospital services or units. On the other hand, several qualitative instruments have been employed to assess the satisfaction of children with their hospital stay, such as interviews [30] and drawings [25], but they fail to allow scalar quantification. The problem is that in order to carry out an evaluation that enables the best practices to be considered in the comparison between hospitals, it is useful to have reliable and valid scales that graduate the children’s satisfaction.

In contrast to the aforementioned studies, the research by Gómez-de-Terreros et al. [31] did develop a questionnaire directly aimed at assessing the satisfaction of pediatric patients with their stay in hospitals in the Andalusian context based on ordinal scale items. These authors designed the Children’s Satisfaction with Hospitalization Questionnaire (CSHQ) composed of 27 items and three content dimensions: staff, infrastructure, and organization. However, its factorial structure has yet to be validated from the psychometric point of view in such a way that reliable scales can be obtained. 

Within this framework, the first objective of this article is to explore the factorial structure of the CSHQ in the Andalusian population to ascertain the underlying dimensions of the satisfaction of children and adolescents with their stay in hospitals. Based on this factor analysis, it is intended to develop valid measurement scales of pediatric patients’ satisfaction that allow comparing and improving hospital practices.

## 2. Method

### 2.1. Participants

The participants were 623 Andalusian minors (339 girls and 267 boys; 17 missing data) ranging in age from 6 to 17 years old (*M* = 10.63 years; *SD* = 2.48) with at least one episode of a three-day hospitalization in an Andalusian public hospital, who had completed the CSHQ. The hospital units in which children had been admitted were: general pediatrics unit (66.0%), oncohematology (7.9%), nephrology (5.5%), surgery (17.8%), and internal medicine (2.8%).

The majority of the children were in hospital for no more than one week with no surgical intervention, but the range includes hospital stays of more than four weeks and numerous surgical interventions (see Table 1 to check the background data of the participants). 

The participants were recruited by hospital teaching staff from 16 public hospitals of the Andalusian Health Service (SAS in Spanish). The teaching staff of the children’s classes in the hospitals selected patients from 6 to 17 years old, hospitalized at least for three days and with adequate reading and understanding levels. These competences are educational objectives at this age in Andalusia. The participants were always accompanied by the teacher in case any clarification was necessary, but they could not make suggestions. In each hospital, those responsible for applying the questionnaire identified the patients at their convenience in the moment of being discharged. They contacted the parents/guardians and the children, and once they had all signed the informed consent form, after having explained the instructions and requested sincerity in their answers, the children answered the questionnaire at the hospital.

All families’ participation was voluntary, once an informed consent form was signed in accordance with the Declaration of Helsinki. The researchers explained the aims of the project and assured participants’ anonymity would be kept under any circumstances. The participants could withdraw without giving any reason. The Coordinating Committee of Ethics of Biomedical Research in Andalusia has approved the study.

Ten of the hospitals were general university hospitals, while six were county hospitals. They were selected with a criterion of accessibility from the network of 29 SAS public hospitals with a classroom and distributed across the eight Andalusian provinces. A minimum quota of 35 participants per province was sought to reach at least the proportion of 10 patients for each item of the questionnaire. This sampling procedure strove to cover the entire Andalusian territory, with an approximate area of influence of 100 square kilometers and a mean population of one million inhabitants per province. The mean number of participants per hospital was 39.94 (*SD* = 23.92). For the population of minors who attended classes in the Andalusian hospitals in 2016 (*N* = 19.659; [32]), the final sample size implies a sampling error of 3.86% with a confidence interval of 95% and provided a subject-to-item ratio of 18:1, which is above the usual recommended criteria [33]. 

### 2.2. Questionnaire 

The CSHQ is a questionnaire developed by Gómez-de-Terreros et al. [31]. It consists of 27 items concerning satisfaction with hospitalization, regarding several aspects of staff, infrastructure, and organization. The items are answered on a five-point scale (from 1 = totally disagree; to 5 = totally agree). Nine of the items expressed a negative assertion about the hospital. The contents of the items were distributed across three content domains: staff (9 items), infrastructure (10 items), and organization (8 items). The questionnaire had been previously tested in a study of 138 pediatric patients and showed an acceptable internal consistency coefficient (Cronbach’s alpha: α = 0.779; [31]).

### 2.3. Statistical Analysis 

After the data collection, the scores of those items with a negative assertion were reversed. The analyses to be carried out to fulfil the main objective of this work were the following sequence:(a)First of all, we implemented descriptive analyses of the kurtosis and skewness of the items, and Mardia’s test [34] to assess the normality of their distribution and decide the most appropriate correlation matrix to be used.(b)Secondly, we performed an analysis of the suitability for factorial analysis of the inter-item correlation matrix, based on the determinant of the matrix, Bartlett’s sphericity test, and the Kaiser–Meyer–Olkin test.(c)Then we developed an optimal implementation of parallel analysis (PA; [35]) based on the minimum rank factor analysis of 500 random correlation matrices obtained by the permutation of the raw data [36] in order to determine the dimensions to be retained in the CSHQ.(d)An exploratory factor analysis (EFA) served to explore the underlying structure of the 27-item CSHQ correlation matrix with unweighted least squares factoring as the extraction method together with promin rotation [37]. The assessment of the goodness of fit was done with the comparative fit index (CFI) and the root mean square error of approximation (RMSEA). We used as criteria their usual cut-offs (CFI > 0.95; RMSR < 0.08; RMSEA < 0.06; [38]) as was Kelley’s criterion of expected mean value of RMSR for an acceptable model (0.045).(e)For the scale obtained from the CSHQ, we assessed the closeness to unidimensionality at the item level [39]. Scales can be treated as essentially unidimensional when a value of UniCo (unidimensional congruence) larger than 0.95, a value of ECV (explained common variance) larger than 0.85, and a value of REAL (residual absolute loadings) lower than 0.300 are achieved [39].(f)After that, we assessed the scale internal consistency using standardized Cronbach’s alpha, McDonald’s omega and the greatest lower bound to reliability, and the scale sensitivity ratio (SR).(g)Finally, in order to assess the discriminative capacity of the scale obtained, we studied its relationship with several variables: gender and age of the patients, number of surgical interventions, duration of stay, hospital, and medical unit.

The statistical analysis was performed using FACTOR version 10.8.02 (Rovira i Virgili University, Tarragona, Spain) [40,41] and SPSS 25.0 (SPSS Inc., Chicago, IL, USA).

## 3. Results

### 3.1. Descriptive Analyses of the Distributional Properties of the Items

An initial analysis of the distribution of responses showed skewness or kurtosis in the majority of the items (considering absolute values equal to or greater than 1.0), where only three items (16, 21, and 23) were assumed as normally distributed (Table 2). The analysis of Mardia’s test [34] for multivariate kurtosis corroborates a significant deviation from normality (*MK* = 90.822; *p* < 0.0001). The mean number of missing responses per item was 7.37 (*SD* = 6.36), ranging from one (item 2) to 33 (item 21) missing elements of data. The majority of the subjects answered all the items (494; 79.3%).

### 3.2. Suitability for Factorial Analysis of the Inter-Item Correlation Matrix 

The inter-item correlation matrix was moderately suitable for factorial analysis, based on the determinant of the matrix (0.0146), Bartlett’s Sphericity test (χ^2^ = 2043.3; *df* = 351; *p* = 0.00001), and the Kaiser–Meyer–Olkin test (KMO = 0.77651). 

### 3.3. Dimensions to Be Retained in the CSHQ

In order to determine the dimensions to be retained in the CSHQ, we carried out an optimal implementation of parallel analysis (PA) [35] based on the minimum rank factor analysis of 500 random correlation matrices obtained by the permutation of the raw data [36]. The PA recommended a one-factor solution. 

### 3.4. Exploratory Factor Analysis and Goodness of Fit

We then tested the goodness of fit of the one-factor solution. The method for factor extraction was that of the robust unweighted least squares from the polychoric inter-item correlation matrix. The robust analysis was Bias-corrected and accelerated (BCa) [42] based on 500 bootstrap samples for the estimation of the asymptotic covariance/variance matrix, and Bootstrap confidence intervals of 95%. The comparative fit index (*CFI* = 0.936), the root mean squared residual (*RMSR* = 0.093), and the root mean square error of approximation (*RMSEA* = 0.065) showed that the one-factor model was underfit for the 27-item questionnaire. All these indices suggest the need for item depuration to achieve an adequate fit. Deletion of items from the 27-item CSHQ was first performed using factor loadings in the EFA. We removed items that presented poor loadings (<0.30) or whose communalities were under 0.50. After item deletion, the factor analysis was then performed a second time to ensure the decision regarding the factor structure of the questionnaire. 

### 3.5. Closeness to Unidimensionality at the Item Level

We subsequently performed a depuration process by deleting the item with the lesser values each time, until the unidimensional criteria and the goodness-of-fit indices were achieved (UniCo = 0.990; ECV = 0.889; REAL = 0.158; *GFI* = 0.991; *RMSR* = 0.0431). The scale finally obtained contains eleven items adjusted to a unidimensional solution that explains 37.8% of the original data variance (Table 3). 

### 3.6. Scale Internal Consistency

The internal consistency indices showed suitable reliability (standardized Cronbach’s alpha = 0.828; McDonald’s omega = 0.831; greatest lower bound to reliability = 0.869). 

This satisfaction scale has a sensitivity ratio (SR) of 2.001, which indicates that two satisfaction levels can be differentiated based on the factor score estimates. The reliability of the scale is well-suited for low and central levels of satisfaction which are those that can be used to detect deficiencies in the hospital practices (Figure 1). The graphic shows the conditional reliabilities, with the cut-off value of 0.80 as a horizontal dotted line, against the factor standardized score estimates (*M* = 0.00; *SD* = 1.00).

### 3.7. Relationship with Other Variables

Finally, we analyzed the relationships between the scale scores with patient and hospital variables. This analysis showed that the gender of the minor (*t* = −0.959; *df* = 542; *p* = 0.338), the number of hospitalizations (*r* = −0.085; *p* = 0.051; *N* = 529), the duration of the last hospitalization (*r* = −0.082; *p* = 0.057; *N* = 540), and the number of surgical interventions (*r* = −0.082; *p* = 0.056; *N* = 540) fail to present significant relationships with the satisfaction of pediatric patients with their hospital stay. By contrast, age (*r* = −0.154; *p* < 0.001; *N* = 543), hospital (*F* = 4.784; *df*_1_ = 15; *df*_2_ = 540; *p* < 0.001; $\eta_{p}^{2}$ = 0.117), and the medical unit (*F* = 5.017; *df*_1_ = 4; *df*_2_ = 505; *p* = 0.001; $\eta_{p}^{2}$ = 0.038) all show significant relationships with the satisfaction of pediatric patients. Specifically, older patients show lower satisfaction than younger patients, and the oncology/hematology units are those that produce the least satisfaction, and surgery gives patients the greatest satisfaction.

## 4. Discussion 

The results obtained in this research showed that the measurement scale obtained from the CSHQ questionnaire is better suited to a unidimensional solution. It seems that the items related to the hosting aspects of the hospital stays have a greater impact on children’s satisfaction than other dimensions (staff or hospital organization). The adjustment of satisfaction items to only one dimension has been previously described in certain studies (e.g., [4]), and other studies have found that environment quality (also named tangible aspects), such as room and bath comfort, cleanliness, temperature, and food service, is the dimension with the highest influence on patient satisfaction [1,5,43,44]. In coherence with these studies, our results show that children seem to express a more holistic view based on the most apparent aspects of their stay in the hospital. It is possible that children can differentiate between elements (staff, infrastructures, organization, etc.), but in the end, they are left with an overall impression. Since satisfaction is related to age [1,4], it is probable that older children could obtain multidimensional results more similar to those of adults. However, a unique scale that enables the evaluation of satisfaction with the hospital stay throughout the age range of pediatric patients is better suited to a one-factor solution.

Nonetheless, satisfaction is a multidimensional construct according to most of the authors [2,3,4,5]. In studies of adult satisfaction [16], or when the parents are those who answer the questionnaires, the emergence of several dimensions is confirmed. In children, however, this does not seem to be the case. This unidimensional view of patients’ satisfaction is partly supported by studies that have found a single second-order dimension that groups the different dimensions of patients’ satisfaction with the hospital, and by those that show that all dimensions of satisfaction are closely related [16,45,46]. As an advantage, the final scale obtained is shorter than the original questionnaire and shows satisfactory levels of internal consistency and adjustment to the unidimensional structure. 

### Limitations and Future Studies

The main limitation of the study is the convenience sampling procedure that does not guarantee the representativeness of the participants. The selection of patients who met the three inclusion criteria—knowing how to read comprehensively, having stayed in the hospital at least three days, and agreeing to participate in the study—was left at the discretion of the collaborating teachers. Unfortunately, we did not ask our collaborators to record the characteristics of those patients who refused to participate in the study, since they had to make their collaboration compatible with their work in the hospital. For this reason, we cannot quantify the rejection rate or evaluate the existence of biases in the sample selection. Thus, it is clear that to finally validate the scale, further steps are needed.

The wide age range of the final sample also limits the conclusions of this study. The adjustment to the characteristics of the pediatric population that attends the Andalusian hospital classrooms, patients between 6 and 17 years old, implies great differences in understanding and perception. Our results suggest that older pediatric patients have a more complex perception and understanding of their hospital stay. These differences make it likely that the scale obtained could be different according to age groups, which should be explored in future studies.

Another limitation of this work is that large city hospitals of Andalusia are represented to a greater extent in the data. These hospitals serve the majority of the population with health problems of a more serious nature that require overnight stays and are usually more saturated than county hospitals. In addition, this usually involves the transfer of the family and the patient to the city, which is a stressful event on its own, regardless of the hospital conditions. Perhaps the needs of the child are different depending on how foreseeable the hospitalization and the severity of the illness are. In our study, children are taken as a single group that has several levels of severity of illness, despite the fact that the unit where they are admitted can have any of a number of repercussions on health (such as oncology vs. traumatology). Neither has the number of hospitalizations been controlled nor has their duration, since the children who answered had at least three days of hospitalization, but no maximum was established; it is therefore foreseeable that the responses can differ widely of those cases in which the durations were much longer.

For future studies, it would be interesting to analyze the relationships of this main dimension of satisfaction with other characteristics of hospitals and pediatric patients, which would enable factors that negatively affect patient satisfaction to be diagnosed and preventative action to be taken. For future applications of the satisfaction questionnaire used herein, it would be interesting to modify its format to make it more attractive to minors—for example, through analogue visual responses instead of Likert-type numbers, as has already been shown in previous work [47].

## 5. Conclusions

A unidimensional scale from the Children’s Satisfaction with Hospitalization Questionnaire (CSHQ) was obtained as a possible valid and reliable instrument for the assessment of the satisfaction of hospitalized children in Andalusia. Starting from a 27-item instrument, and after its administration to a total of 623 children from different hospitals in Andalusia, the statistical analysis suggests that the original dimensions are significant for only one single factor. Eleven items that refer to the quality of the hosting aspects of hospitals (or tangible aspects) represent this factor. Other studies have also highlighted the importance of such elements, especially when dealing with children.

In spite of this main conclusion, and due to the information that can be provided by the rest of the items in the questionnaire, it is interesting to administer the entire questionnaire to collect qualitative information about a complex emotion such as satisfaction. It would be interesting in future validation steps of the obtained scale to update the pediatric samples to the characteristics of the child population.

The interest of obtaining a scale that validly and reliably orders the satisfaction of pediatric patients is that it allows a more accurate evaluation of the conditions that affect them. In this way, it would be possible to detect those situations that are more deficient, to improve them by comparing hospital practices. In this way, hospitals could focus on those aspects that really have an impact on the perceived well-being of their users, improving the economic efficiency of the changes that pursue that aspect. While satisfaction can be considered a complementary aspect of the quality of medical care, there is a broad consensus that it can condition the therapeutic success of treatments administered in hospitals. For this reason, it is important to have instruments that allow a valid assessment of pediatric patient satisfaction.

## Figures and Tables

**Figure 1 ijerph-16-03110-f001:**
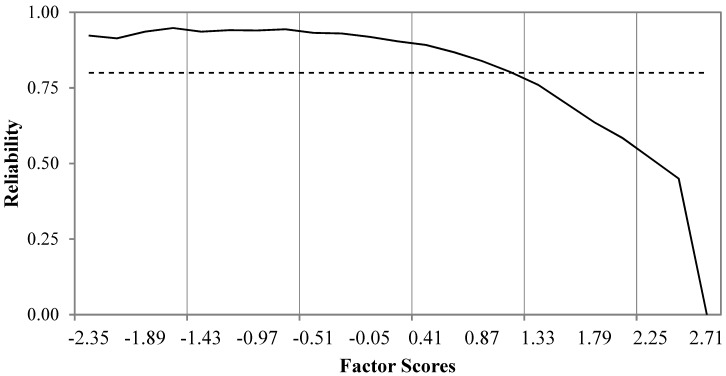
Distribution of conditional Bayes expected a posteriori estimation (EAP)/Orion reliabilities for the factor scores of the unidimensional satisfaction scale obtained from the CSHQ.

**Table 1 ijerph-16-03110-t001:** Background data of the participants.

Variables	Statistics		Missing
Male/Female ratio (%)	267/339	(44.1/55.9)	17
Mean Age (SD)	10.63	(2.48)	17
Duration of Hospitalization (%)			22
Between 3 and 7 days	341	(56.7)	
Between 8 and 14 days	151	(25.1)	
Between 15 and 21 days	51	(8.5)	
Between 22 and 28 days	14	(2.3)	
29 days or more	44	(7.3)	
Number of surgical interventions (%)			21
None	300	(49.8)	
One	178	(29.6)	
Two	43	(7.1)	
Three or more	81	(13.5)	

**Table 2 ijerph-16-03110-t002:** Univariate descriptive statistics for the Children’s Satisfaction with Hospitalization Questionnaire (CSHQ) items.

Item	Mean	Variance	Skewness	Kurtosis	Item-Total *r*	Missing
1	4.794	0.257	−3.006	11.475	0.325	3
2	4.757	0.261	−2.117	4.143	0.342	1
3	4.909	0.123	−4.685	26.029	0.210	10
4	4.526	0.585	−1.679	2.705	0.355	5
5	4.350	0.734	−1.359	1.773	0.291	5
6	3.279	1.910	−0.173	−1.126	0.256	5
7	4.445	0.862	−2.089	4.339	0.375	2
8	4.538	0.751	−2.252	5.090	0.293	5
9	4.759	0.369	−3.187	12.341	0.300	7
10	4.702	0.569	−3.130	10.408	0.191	12
11	4.069	1.299	−1.196	0.652	0.191	9
12	4.690	0.720	−3.028	8.649	0.316	9
13	4.587	1.052	−2.513	5.152	0.252	7
14	4.455	0.997	−2.047	3.667	0.194	4
15	4.585	0.729	−2.399	5.717	0.399	6
16	3.822	2.572	−0.897	−0.903	0.221	11
17	4.656	0.720	−2.867	8.181	0.232	5
18	4.000	1.490	−1.057	0.085	0.395	6
19	4.235	1.520	−1.467	0.879	0.348	4
20	4.439	1.230	−1.938	2.539	0.392	9
21	3.735	2.069	−0.702	−0.920	0.316	33
22	4.194	1.509	−1.360	0.678	0.279	18
23	3.674	2.382	−0.766	−0.959	0.303	3
24	4.267	1.471	−1.546	1.212	0.426	11
25	4.324	1.430	−1.712	1.734	0.326	3
26	4.518	0.812	−2.290	5.314	0.401	3
27	4.540	0.682	−2.130	4.800	0.449	3

**Table 3 ijerph-16-03110-t003:** Items included in the unidimensional scale ordered by factor loading.

Items	Loading
My room has been comfortable	0.739
My stay in the hospital was well	0.696
The hospital is ugly ^a^	0.628
My hospital room has had everything I needed (e.g., TV)	0.647
There have been things in the hospital so I would not be bored	0.632
My room was dirty ^a^	0.535
The hospital is well maintained	0.534
I have slept well without being disturbed	0.472
Porters moving me through the hospital have been kind	0.461
I liked the age of the children in my room	0.422
Hospital food seemed bad to me ^a^	0.317

^a^ Note: The scoring of these items was reversed.

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
