# Peer review of "A Measurement Scale to Assess Children’s Satisfaction with Hospitalization in the Andalusian Population"

_ijerph, 2019, doi:10.3390/ijerph16173110_

Round 1

Reviewer 1 Report

 It is clear that the validation of the scale has many limitations, as stated in the manuscript, such as the number of days the child remains admitted (a wide range of days), the severity of the pathology that causes admission or age (from 6 to 17 years old); So this study could be a first phase of validation of the CSHQ questionnaire, since there are different points to be polished.

 It´s true that satisfaction is a complex emotion, and in a child even more, that is why it would be interesting to define more the scale focused on this population by updating it to the characteristics of the child population in a future second validation step; because currently said validation of the questionnaire would be incomplete and this point would have to be reflected in some way in the article

 It is important to further develop the conclusions and implications for Public Health in the pediatric area of this evaluation method

Author Response

Thank you for your comments they have helped us to improve the quality of the manuscript. We have edited the paper according to the comments of the reviewers using the "Change tracking" function in Microsoft Word.

RESPONSES TO REVIEWER 1

Point 1.It is clear that the validation of the scale has many limitations, as stated in the manuscript, such as the number of days the child remains admitted (a wide range of days), the severity of the pathology that causes admission or age (from 6 to 17 years old); So this study could be a first phase of validation of the CSHQ questionnaire, since there are different points to be polished.”

Response 1.

We consider that the validation of an instrument for the assessment of psychosocial aspects, such as satisfaction, is a process rather than an end product. In that sense, we agree that this work is a first step in the validation of a scale for the assessment of the satisfaction of pediatric patients in the Andalusian hospital environment.

In any case, it is convenient to differentiate the CSHQ questionnaire from the one-dimensional measurement scale that we have extracted from it. The questionnaire as a whole is an informative instrument, which allows registering the ratings of the patients on different aspects of their hospital stay -staff, infrastructure and organization-. On the other hand, the Scale obtained in this work aims to be an evaluative instrument that validly order the different levels of satisfaction presented by patients. This work deals only with the process of obtaining such a scale, but not with the validation of the CSHQ questionnaire. For this reason, we have considered that in order to conform to the reviewer's recommendations, it is more appropriate to title the work as "A measurement scale ...", rather than validation. We have also tried to improve the introduction in this regard. We have reordered the arguments, lines 47 to 54, and changed the wording of the objective, lines 61 to 65.

However, we do not think that the variety of circumstances reflected in our sample of pediatric patients - age, severity of their pathologies, number of days of hospital stay, etc. - is actually a problem. Precisely this variety was the one sought to be able to represent the largest possible number of situations faced in these hospitals with pediatric patients.

Point 2. “It´s true that satisfaction is a complex emotion, and in a child even more, that is why it would be interesting to define more the scale focused on this population by updating it to the characteristics of the child population in a future second validation step; because currently said validation of the questionnaire would be incomplete and this point would have to be reflected in some way in the article.”

Response 2.

As we have indicated in the previous answer, the work tries to obtain a measurement scale that allows ordering in a valid and reliable way the satisfaction of pediatric patients in Andalusian hospitals. This is not about the validation of the whole questionnaire. We agree with the reviewer that in subsequent works for the validation of the scale we must adjust the samples to better represent the characteristics of the pediatric population.

In the conclusions of the work, lines 265 to 273, we have included a clarification in this regard.

Point 3. It is important to further develop the conclusions and implications for Public Health in the pediatric area of this evaluation method.”

Response 3.

Precisely, the interest of obtaining a measurement scale that validly and reliably orders the satisfaction of pediatric patients is that it allows a more precise evaluation of the conditions that satisfy their patients. In this way, it would be possible to detect those situations that are more deficient, in order to improve them by comparing hospital practices. Such data allow the hospitals to focus on those aspects that have an impact on the perceived well-being of their users. Although satisfaction can be considered a complementary aspect of the quality of health care, however there is a broad consensus that it can condition the therapeutic success of the treatments given in hospitals, especially with children. For this reason, it is important to have instruments that allow valid evaluation of pediatric patient satisfaction.

Responding to the reviewer's comments, we have added this argumentation in the conclusions of the work, lines 265 to 273. We have also changed the explanations that accompany Figure 1, lines 182 to 187, to highlight that the results indicate that the scale evaluates in a more reliable way the low and central levels of satisfaction with hospital stay, which are those that can be used to detect deficiencies in the hospital practices.

Reviewer 2 Report

I have enjoyed reading this manuscript. It is in good shape and easy to read. There are a few comments that I would like to raise, on the content and the analysis:

Topic

The topic is very interesting. Patient satisfaction is getting more and more important for health care provision and also for researchers. Therefore, the topic is very well chosen. Also because the analysis is focused on minors, who many times do not answer by themselves about their opinions.

Sample

With respect to the sample and its collection: how were children collected? I can see in lines 74 to 92 some information. However, I do not find information on the children who did not complete the survey or why they did not do it. I would need to know a bit more on this: was the survey presented to just a sample of children meeting the criteria? how was this sample obtained? randomly? Did it depend on the hospital teaching staff? Or how many children suffered hospitalizations of more than 3 days in Andalusia in the year of the study? In my opinion this is important because there might be a selection bias in the sample. If so, the results would not be valid.

Methods

Related to the methods, I find very interesting the analysis of the unidimensional scale. However, I have a major comment with the design of the survey: out of the 27 items, all of them refer either to staff, organization or infrastructure. None has to do with the performance of the health provision. My problem is that I would think that performance is also important to create the opinion of patients (even if they are children). I understand that sample is what it is, and now it is too late to add questions. However, something that it is feasible is to add to the analysis some control variables such as the diagnosis, or the level of severity of children. Maybe there is some correlation between their health status and their opinion about the hospital, and that might not necessarily be related to the quality of the service provision. Or maybe some hospitals, dealing with "easier" patients (less severe) perform better because they may be focus on other issues. I would like to know whether results are different by type or size of hospital (general vs county). This is very related to one of the limitations stated in the manuscript.

Discussion and conclusion

I find also very interesting the comment on the difference between unidimensional satisfaction for children and multidimensional satisfaction for adults. I am not sure whether it is the design of the study what drives this result. Were surveys to adults similar to this one? or in their questionnaires, were there questions related to their diagnosis, severity or performance in the provision?

In general, I think the manuscript provides a nice view on satisfaction in children, but as mentioned in the last paragraph, results might be conditioned by the pathologies or severity, and that might be easy to add here. 

Author Response

Thank you for your comments they have helped us to improve the quality of the manuscript. We have edited the paper according to the comments of the reviewers using the "Change tracking" function in Microsoft Word.

RESPONSES TO REVIEWER 2

Point 1. With respect to the sample and its collection: how were children collected? I can see in lines 74 to 92 some information. However, I do not find information on the children who did not complete the survey or why they did not do it. I would need to know a bit more on this: was the survey presented to just a sample of children meeting the criteria? how was this sample obtained? randomly? Did it depend on the hospital teaching staff?” 

Response 1: These comments are quite appropriate, since they affect the representativeness of the sample, which is one of the aspects that have most concerned us in the development of this work. In our document we had not made clear the procedure for the selection of pediatric patients. The sample was obtained for convenience due to the limitations inherent to a study of this nature, in which the teaching staff of the hospital classrooms were voluntaries according to their availability. Although not using a random sample limits the representativeness of the study, however it has been guaranteed that the entire Andalusian territory was covered. This was done by imposing a minimum quota of pediatric patients for each of the provinces that cover the Andalusian territory, leaving at the discretion of the collaborating teachers the selection of patients who met the three inclusion criteria: knowing how to read comprehensively, having stayed in the hospital at least three days and agreeing to participate in the study. Unfortunately, we did not ask our collaborators to registry the characteristics of those patients who refused to participate in the study, since their collaboration was made compatible with their work in the hospital. For this reason we cannot quantify the rejection rate or evaluate the existence of biases in the sample selection. However, as we have answered another reviewer, due to the variety of the sample regarding the Andalusian provinces, age of the patients, duration of their stay, number of surgical interventions, hospitals and medical units, we are confident that our data adequately illustrate most of the situations that are faced in Andalusian public hospitals with pediatric patients. However, it is true that it is more in line with the characteristics of our study to remove the term "Validation" of the title, which is more appropriate for random samples.

We have implemented this change in the title of the work, and we have also reviewed some of the statements of the work in this regard. We have also reordered the explanations of the selection procedure, lines 79 to 88, in order to clarify these aspects. We have added a paragraph in the limitations section to expose these concerns, lines 226 to 233.

Point 2. “How many children suffered hospitalizations of more than 3 days in Andalusia in the year of the study? In my opinion this is important because there might be a selection bias in the sample. If so, the results would not be valid.”

Response 2: Unfortunately there are no official statistics for the duration of stay of pediatric patients in Andalusian hospitals. The last report published by the Andalusian government (https://www.juntadeandalucia.es/organismos/saludyfamilias/consejeria/sobre-consejeria/estadistica-cartografia/paginas/estadisticas-hospitalarias.html), only includes the total numbers of pediatric patients admitted to Andalusian hospitals. In 2016 there were a total of 219,760 hospitalized pediatric patients, of which only 19,659 attended hospital classrooms. But we cannot quantify the population who has remained in Andalusian hospitals for three or more days. Nonetheless, it is clear that our study cannot defend the representativeness of the sample in statistical terms with respect to the total pediatric Andalusian patients. But the objective of the work is to obtain a scale of measurement that allows comparative analysis of the practices of Andalusian hospitals that affect the satisfaction of their pediatric patients. For this purpose, we understand that the criteria used to select the sample remain valid. On the one hand it is necessary that children and adolescents have experienced their stay in the hospital at least enough time to properly assess the different aspects that are included in the CHSQ. For this reason, limiting the sample to those children who have spent at least three days in the hospital still seems appropriate. It is also likely that there have been selection biases by the collaborating teachers of the study. But when we consider the group of hospitals included in the study, the territory covered, the diversity of ages and situations of the patients reflected in the sample, we believe that they are quite illustrative of the characteristics of the population to which the measurement scale is directed. In any case, we agree that these possible biases prevent us from speaking appropriately of validation of the scale, which we have corrected in the title of the work and in the text of the document, as we have explained in the previous response.

Point 3. “Related to the methods, I find very interesting the analysis of the unidimensional scale. However, I have a major comment with the design of the survey: out of the 27 items, all of them refer either to staff, organization or infrastructure. None has to do with the performance of the health provision. My problem is that I would think that performance is also important to create the opinion of patients (even if they are children). I understand that sample is what it is, and now it is too late to add questions. However, something that it is feasible is to add to the analysis some control variables such as the diagnosis, or the level of severity of children. Maybe there is some correlation between their health status and their opinion about the hospital, and that might not necessarily be related to the quality of the service provision. Or maybe some hospitals, dealing with "easier" patients (less severe) perform better because they may be focus on other issues. I would like to know whether results are different by type or size of hospital (general vs county). This is very related to one of the limitations stated in the manuscript.”

Response 3: This is a very interesting comment for us, since we have taken it into account during the development of this work. We have tried to include reflections related to this aspect in the sections of future studies and conclusions, but we will not have made enough emphasis. We have now added a new paragraph in the conclusions, lines 265 to 273, aimed at clarifying these aspects better.

            It should be noted that we understand that patient satisfaction is a complementary aspect of the quality of health care, but it is not the central one. For us it is clear that, as the reviewer comments, the quality of the health service offered plays a decisive role. However, there is some consensus in considering that the satisfaction perceived by patients can interfere with the effectiveness of health treatments. We think is especially true in pediatric patients, in which emotional aspects can determinate more easily their assessment of situations. Our study shows that in these pediatric patients, their satisfaction seems to be quite conditioned by the most apparent aspects of hospital stay, such as all features related to the hosting offered by hospitals.

            Unfortunately, our data does not allow us to answer all the questions raised by the reviewer, such as the relationship of patient satisfaction with the severity of their pathology. Yes, we have found that there are significant differences in the satisfaction of minors according to the hospital, but these differences do not appear related to the type of hospital, as we have reflected in the results section. As the reviewer points out, this is related to some of the limitations we stated, as well as in the proposals for future developments.

Point 4. “I find also very interesting the comment on the difference between unidimensional satisfaction for children and multidimensional satisfaction for adults. I am not sure whether it is the design of the study what drives this result. Were surveys to adults similar to this one? or in their questionnaires, were there questions related to their diagnosis, severity or performance in the provision?. In general, I think the manuscript provides a nice view on satisfaction in children, but as mentioned in the last paragraph, results might be conditioned by the pathologies or severity, and that might be easy to add here.”

Response 4: Although our initial hypothesis is that the satisfaction of pediatric patients would be structured in the three dimensions included in the CHSQ questionnaire, however, the results did not fit that multidimensional solution. The most adjusted solution turned out to be the unidimensional one, but a depuration process was necessary to select those items that offered adequate indexes of fit as well as good reliability indexes. These are important aspects for the objective of the study, which was to obtain scales of measurement of patients’ satisfaction for adequate comparison of hospital practices.

As we have already pointed out in our previous response, we cannot analyze the relationship of satisfaction with the severity of patient pathologies. Yes, we found a relationship between patient satisfaction and age, so that older patients tended to have a lower score on the one-dimensional scale referring to the hosting aspects of hospitalization. In that sense, we agree with the reviewer that older patients are likely to show a multidimensional perspective of their satisfaction, as is the case with studies with adult patients that we have pointed out in our review.

Reviewer 3 Report

Sincere thanks for your interesting manuscript.

There are many satisfaction studies with health services. But there are fewer studies to evaluate the satisfaction and perceived quality of hospitalized children.

A major problem is the age of hospitalized children, the ability they may have to respond to their assessment of hospital stay, based on previous expectations they may have of what it means to be hospitalized.

This study tries to validate the Childrens´s Satisfaction with Hospitalization Questionnaire (CSHQ) in the Andalusian hospital context, there is already a previous pilot and validation. There are some issues that should be clarified to better understand the manuscript:

Introduction:

A review of the scales or methods that currently exist to assess health services from a pediatric point of view is made, even traditionally the parents of these children have been the people who have valued the services and this study or the scale that is The validation process is designed to be answered by the patients themselves. A clear exposition of the problem is made, and there is a scale applied to the pediatric population, but it is necessary to adapt it to the context of Andalusian hospitals

Methods:

2.1 Participants

Line 88. There is a description of the sample ratio, but not a justification of the sample size, could it be described?

Line 93. There are differences in understanding and perception between a 6 year old and a 17 year old. Has this factor been taken into account when the results have been analyzed?

2.3 Statistical analysis

Appropriate methods and indices are used to carry out the factor analysis and the consistency of the scale, but would it be possible to explain them in an orderly manner and then explain the results in this order?

Results:

Line 144. Table 2. Is it possible to put the sentences of the 27 items?

Line 145-151. The factor analysis, could you see the matrix of main components to see how the items are grouped into a single dimension and the rest of the items how they are grouped? Being a manuscript on the validation of a model, it is important to visually see how the model looks (even this article would add value to the corresponding structural equation, but only as a suggestion in this case)

Line 170. Table 3 only describes the 11 items that are part of the single factor factor solution, and the rest of the items?

Discussion

In the manuscript of Gómez de Terreros (2016) in the Gaceta Sanitaria Journal, a 3-dimensional questionnaire with 27 items is developed to assess satisfaction with hospital stay in children and young people. It should be explained since they are the same authors, why this article strengthens the idea that a single dimension is better? What is the add value for this study? Could this second validation be better defended?

Line 233-234. A limitation is the age groups, it is clear that the easiest thing to assess is what is seen, which is tangible but perhaps by age groups the scale could be different and even incorporate different formats according to the age groups.

References

References must be numbered in order of appearance in the text (including table captions and figure legends) and listed individually at the end of the manuscript. En el manuscrito se sigue el estilo APA y no es el formato de esta revista.

Author Response

Thank you for your comments they have helped us to improve the quality of the manuscript. We have edited the paper according to the comments of the reviewers using the "Change tracking" function in Microsoft Word.

RESPONSES TO REVIEWER 3

Point 1. Line 88. There is a description of the sample ratio, but not a justification of the sample size, could it be described?”

Response 1: It is true that we have not justified the sample size, and that we have only described the proportion of error regarding the population of hospitalized minors in Andalusia that attend hospital classrooms. For greater clarity, we have now added in the text the population size of pediatric patients who were treated in hospital classrooms during 2016 when data collection was carried out (line 96).

But as we have indicated in our responses to other reviewers, our study cannot justify the representativeness of the sample in statistical terms. After analyzing the comments of the reviewers, we have understood that it is a confusion to refer to the work as "validation of a scale", therefore we have changed it in the title and in the document. Our research is about obtaining measurement scales from the CHSQ questionnaire, based on an Exploratory Factor Analysis. For this purpose the relevant question is that the sample of participants reflects the greatest possible variety of situations that are faced in Andalusian public hospitals with respect to their pediatric patients and that a minimum proportion of 10 participants is met for each item of the questionnaire (33), which would involve at least 270 participants. In this sense, we understand that our sample adequately represent this variety while covering all the provinces of the Andalusian territory, and exceed the proportion indicated above.

The explanation of the final sample size obtained is made between lines 89 and 98 of the text. We have added in the text that a minimum quota of 35 participants per province was sought to reach at least the proportion of 10 patients for each item of the questionnaire, although it was finally exceeded. In relation to this comment, we have also added an explanatory paragraph in the Limitations section (lines 226 to 233).

Point 2. Line 93. There are differences in understanding and perception between a 6 year old and a 17 year old. Has this factor been taken into account when the results have been analyzed?”

Response 2: We agree with this comment, and we have included in the text an argument related to it (lines 192 to 201). We have taken this into account when analyzing the results on the satisfaction scale according to the age of the patients (line 197), finding a significant negative relationship between them. In this way, older patients show less satisfaction with their hospital stay depending on the hosting aspects that the final scale includes. We understand that this result comes to support the idea that older pediatric patients have a different perception and understanding of satisfaction with their stay, and that this is probably more complex. But we consider that it is appropriate for our study to have used such a wide age range despite its possible differences in understanding. This is how we really adjust to the characteristics of the pediatric population that attends the hospital classrooms, and therefore share conditions regarding their hospitalization.

Point 3. Appropriate methods and indices are used to carry out the factor analysis and the consistency of the scale, but would it be possible to explain them in an orderly manner and then explain the results in this order?”

Response 3: The reviewer is right in this regard. We have tried to reorder the analyses and the results, so that they are clearer. We have numbered with letters from a to g the analyses and the results to make them more understandable.         

Point 4. Line 144. Table 2. Is it possible to put the sentences of the 27 items?”

Response 4: Table 2 is oriented to present several descriptive statistics of each of the CHSQ items. For this reason, it does not seem appropriate to also include the text of each of the items, since the table would end up being too loaded with content that would make it difficult to read. We understand that the reader interested in knowing the full content of the CHSQ items can always turn to the original work (31) to read the items. Of course, if you consider that it is apropiase to include the questionnaire, we can do it perhaps as an annex.

Point 5. Line 145-151. The factor analysis, could you see the matrix of main components to see how the items are grouped into a single dimension and the rest of the items how they are grouped? Being a manuscript on the validation of a model, it is important to visually see how the model looks (even this article would add value to the corresponding structural equation, but only as a suggestion in this case)”

Line 170. Table 3 only describes the 11 items that are part of the single factor factor solution, and the rest of the items?”

Response 5: The responses to these comments are closely related. The Exploratory Factor Analysis led us to reject the multidimensional solution that initially seemed more appropriate, recommending a one-dimensional solution as best suited to the responses of these pediatric patients. From that moment on, the rest of the analysis was done to fit the model of a single factor. For this reason, in table 3 only eleven items appear, those with a minimum load on the factor (lines 171-173) and that were selected for meeting minimum criteria of one-dimensionality (lines 173 to 175).

We agree that if the model had been adjusted to a three-factor solution as expected, its graphic representation would have provided a very useful tool. But since it is finally a single factor model, we understand that its visual representation is not so necessary.

Point 6. In the manuscript of Gómez de Terreros (2016) in the Gaceta Sanitaria Journal, a 3-dimensional questionnaire with 27 items is developed to assess satisfaction with hospital stay in children and young people. It should be explained since they are the same authors, why this article strengthens the idea that a single dimension is better? What is the add value for this study? Could this second validation be better defended?”

Response 6: It is true that the title and the initial writing of the document presented this work as the validation of the structure of the CHSQ questionnaire. Actually, what is presented here is a scale of measurement obtained from this questionnaire, whose purpose is above all practical. This measurement scale should allow the comparison and improvement of hospital practices that affect the satisfaction of their pediatric patients. For that purpose it is necessary that the scale meets minimum criteria of validity and reliability, and that is what this study developed.

In our work (31) we explained the development of the CHSQ questionnaire along with a pilot study. But that work did not validate the supposed multidimensional structure of the questionnaire, which reflected the three contents included in it - personnel, infrastructure and organization. That was the hypothesis from which we started in this study, but we had to reject it after the exploratory factor analysis. What we can say now is that according to our analyses the multidimensional structure of hospital satisfaction cannot be defended for pediatric patients, at least not in a generalized way. On the contrary, what our data show is that the satisfaction of all these patients with their hospital stay is better suited to a single dimension that reflects the most apparent aspects of the hospital.

Point 7. Line 233-234. A limitation is the age groups, it is clear that the easiest thing to assess is what is seen, which is tangible but perhaps by age groups the scale could be different and even incorporate different formats according to the age groups.”

Response 7: We have responded to another reviewer who makes a comment similar to this, including in the text an argument related to it (lines 213 to 216). We have taken this into account when analyzing the results on the satisfaction scale according to the age of the patients (line 197), finding a significant negative relationship between them. In this way, older patients show less satisfaction with the hosting aspects of their hospital stay. We understand that this results comes to support the idea that older pediatric patients have a different perception and understanding of satisfaction, and that this is probably more complex. But we consider that it is appropriate for our study to have used such a wide age range despite its possible differences in understanding. This is how we really adjust to the characteristics of the pediatric population that attends the hospital classrooms in Andalusia, and therefore share conditions regarding their hospitalization.

We have also taken into account the comment on the change of formats of the questionnaire (lines 248-251).

Point 8. References must be numbered in order of appearance in the text (including table captions and figure legends) and listed individually at the end of the manuscript. En el manuscrito se sigue el estilo APA y no es el formato de esta revista.”

Response 8: We have changed the references style to Vancouver.